

# Long-range magnetic order in the $\tilde{S} = 1/2$ triangular lattice antiferromagnet KCeS$_2$

Gaël Bastien[1], Bastian Rubrecht[1,2], Ellen Haeussler[3], Philipp Schlender[3], Ziba Zangeneh[1], Stanislav Avdoshenko[1], Rajib Sarkar[2], Alexey Alfonsov[1], Sven Luther[2,4], Yevhen A. Onykiienko[2], Helen C. Walker[5], Hannes Kühne[4], Vadim Grinenko[1,2], Zurab Guguchia[6], Vladislav Kataev[1], Hans-Henning Klauss[2], Liviu Hozoi[1], Jeroen van den Brink[1,7], Dmytro S. Inosov[7], Bernd Büchner[1,7], Anja U. B. Wolter[1] and Thomas Doert[3]

**1** Leibniz-Institut für Festkörper- und Werkstoffforschung (IFW) Dresden, 01171 Dresden, Germany
**2** Institut für Festkörper- und Materialphysik, Technische Universität Dresden, 01062 Dresden, Germany
**3** Fakultät für Chemie und Lebensmittelchemie, Technische Universität Dresden, 01062 Dresden, Germany
**4** Hochfeld-Magnetlabor Dresden (HLD-EMFL), Helmholtz-Zentrum Dresden-Rossendorf, 01328 Dresden, Germany
**5** ISIS Neutron and Muon Source, Rutherford Appleton Laboratory, Chilton, Didcot OX11 OQX, United Kingdom
**6** Laboratory for Muon Spin Spectroscopy, Paul Scherrer Institute, CH-5232 Villigen PSI, Switzerland
**7** Institut für Festkörper- und Materialphysik and Würzburg-Dresden Cluster of Excellence ct.qmat, Technische Universität Dresden, 01062 Dresden, Germany

## Abstract

Recently, several putative quantum spin liquid (QSL) states were discovered in $\tilde{S} = 1/2$ rare-earth based triangular-lattice antiferromagnets (TLAF) with the delafossite structure. In order to elucidate the conditions for a QSL to arise, we report here the discovery of a long-range magnetic order in the Ce-based TLAF KCeS$_2$ below $T_N = 0.38$ K, despite the same delafossite structure. Finally, combining various experimental and computational methods, we characterize the crystal electric field scheme, the magnetic anisotropy and the magnetic ground state of KCeS$_2$.

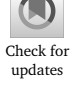

# 1  Introduction

Magnetic frustration is at interest because it can lead to the competition of a large variety of emergent states of broken magnetic symmetry and the possible realization of quantum spin liquid (QSL) states [1–3]. In particular materials with a QSL ground state have caused an abiding fascination within the scientific community due to its strong quantum entanglement, while long-range magnetic order is absent down to zero temperature. This exotic spin-disordered state displays fractionalized quasiparticle excitations relevant for topological quantum computation [4]. QSL states are typically predicted to occur in geometrically frustrated magnets such as triangular, kagome, and pyrochlore lattices [2], and display a macroscopic degeneracy that stabilizes a topologically ordered ground state.

The triangular-lattice antiferromagnet (TLAF) is the simplest case of geometrical magnetic frustration and was comprehensively investigated with $3d$ transition metals as the magnetic ion [3, 5, 6]. DMRG calculations and Monte Carlo calculations determined the 120° phase as the ground state of the Heisenberg spin 1/2 TLAF [7, 8], and this state was observed in several $3d$-TLAF, such as $Ba_3CoSb_2O_9$ [6]. However, a second nearest-neighbor interaction $J_2$ can destabilize the 120° spin-ordered phase into a quantum spin liquid state as shown by DMRG [9, 10] and variational Monte Carlo calculations [11]. In addition, anisotropic magnetic interactions have also been proposed as a way to induce new magnetically ordered or disordered states, such as a QSL state in TLAF [12–17]. The anisotropic component of the nearest-neighbor magnetic exchange interactions can be described by two additional terms with the magnetic interactions $J_{z\pm}$ and $J_{\pm\pm}$ in the Hamiltonian [13]. Recent variational Monte Carlo studies and DMRG studies have shown that the off-diagonal nearest-neighbor interaction $J_{z\pm}$ can lead to the formation of a QSL, while the $J_{\pm\pm}$ interaction suppresses QSL towards a stripe-type magnetic ordered state [15–17].

Anisotropic magnetic interactions can be realized by choosing a magnetic ion with a strong spin-orbit coupling (SOC) such as those of the rare-earth elements. For the two magnetic ions $Ce^{3+}$ or $Yb^{3+}$, with electronic configurations of $4f^1$ and $4f^{13}$, respectively, a $\tilde{S} = 1/2$ spin state can be achieved at low temperature due to the spin-orbit coupling in combination with the depopulation of higher-energy crystal electric field (CEF) levels [13, 14, 18, 19]. Numerous Yb-based TLAF were recently reported and proposed to host a quantum spin liquid ground state: $YbMgGaO_4$ [13, 20], $NaYbO_2$ [21–25], $NaYbS_2$ [19, 21, 26], $NaYbSe_2$ [21, 27, 28], $KYbS_2$ [29],

CsYbSe$_2$ [30] and Yb(BaBO$_3$)$_3$ [31]. In addition, a putative QSL state was recently reported in the Ce-based TLAF CsCeSe$_2$ [30]. While the absence of long-range magnetic order for several of these compounds [13, 20, 27] was associated with the presence of highly anisotropic spin couplings, an analysis of effective superexchange models suggests on the contrary a rather isotropic Heisenberg behavior of the magnetic interactions in Yb-based magnets with Yb ions in cubic or approximate cubic environment [32]. Second nearest-neighbor interactions were also proposed as a possible origin of the QSL state [33] and at this time the source of the quantum spin liquid state in the rare-earth based TLAFs remains still under debate [3].

A way to clarify the origin of the QSL state would be finding a way to tune rare-earth based TLAF from the putative QSL state towards long-range magnetic order. Here, we introduce the Ce-based TLAF KCeS$_2$ which yields magnetic order despite the same delafossite crystal structure [34] and similar composition as the QSL candidates NaYbO$_2$ [21], NaYbS$_2$ [19, 21], NaYbSe$_2$ [21], KYbS$_2$ [29], CsYbSe$_2$ [30] and CsCeSe$_2$ [30]. We report the single crystal growth of KCeS$_2$ by the modified Fujinos method [35], magnetization, electron spin resonance (ESR) and inelastic neutron scattering (INS), along with *ab initio* quantum chemical calculations, as well as low-temperature specific heat, and muon spin spectroscopy ($\mu$SR) measurements. A well separated lowest energy CEF doublet was identified by INS and *ab initio* computations, ensuring the realization of a $\tilde{S} = 1/2$ ground state. Furthermore, a strong easy-plane magnetic anisotropy of KCeS$_2$ is reported and characterized via magnetization measurements up to 30 T, ESR measurements, and electronic-structure calculations, and is interpreted in terms of a strong $g$-factor anisotropy of the ground state. The magnetic ordering at $T_N = 0.38$ K was characterized by specific heat and $\mu$SR experiments. The anisotropic magnetic field-temperature phase diagram $H$-$T$ was established for two inequivalent directions within the basal plane $ab$. This reveals an in-plane anisotropy, which may indicate anisotropic magnetic interactions in KCeS$_2$.

## 2 Crystal growth and structural analyses

Crystals of KCeS$_2$ were grown by the Fujinos [35] procedure with slight modifications. Potassium carbonate (5528 mg, 40 mmol) and cerium dioxide (334.2 mg, 2 mmol) are mixed and thoroughly ground in a porcelain mortar under ambient conditions. A glassy carbon crucible is filled with this mixture and placed in the middle of a ceramic tube in a tube furnace. Before heating up to the target temperature, the whole apparatus including a 1 L flask as CS$_2$ reservoir was flushed with argon (5 L/h) for 30 min. The mixture is heated up to 1050 °C within 3 hours under an unloaded stream of argon (2 L/h). While dwelling one hour, a stream of argon (5 L/h) was used to carry CS$_2$ into the reaction zone to enable the sulfidation. Finally, the furnace was cooled down to 600 °C within 6 hours and then, without further control of the temperature, down to ambient temperature under a slight argon stream ($\approx$ 2 L/h). The CS$_2$ consumption amounted to approximately 20 ml ($\approx$ 0.33 mol) during the whole procedure. The solidified melt was leached with water and the insoluble KCeS$_2$ was filtered-off and washed with water and ethanol. According to the x-ray powder diffractogram, the respective LeBail fit and the comparison with literature data (Fig. 1), KCeS$_2$ was obtained phase pure; no evidence for impurities was found. The product was mainly found as agglomerated and intergrown crystals with dimensions ranging from 0.02 to 2 mm. A few single crystals were found isolated as hexagonally shaped platelets or hexagonal antiprisms. While one of the smaller isolated crystals was chosen for structure analyses, relatively large hexagonal platelets were used for magnetization, specific heat and ESR measurements. A collection of crystals with a total mass of 12 mg coaligned along the $c$ axis was prepared for magnetization measurements in pulsed magnetic fields, and larger collections of non-oriented crystals of 2 g and 300 mg

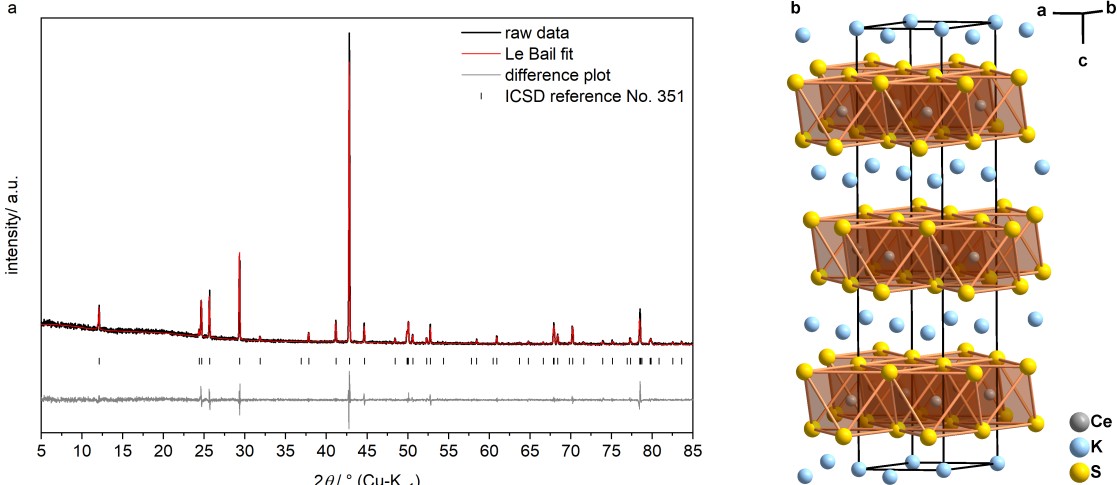

Figure 1: (left) X-ray powder diffraction (solid black line) of KCeS$_2$ at ambient temperature. The red dots represent the Le Bail fit and the solid grey line the difference between experimental and calculated data. The R values of the Le Bail fit are: $R_p =$ 8.38, $w_{Rp} = 11.25$, (GOF = 1.30). (right) Crystal structure of KCeS$_2$ consisting of layers of edge sharing CeS$_6$-octahedra alternating with layers of K$^+$.

were used in our INS and $\mu$SR experiments, respectively.

Single crystals of the nonmagnetic analog KLaS$_2$ were grown using the same procedure to serve as reference for the phonon contribution to the specific heat.

KCeS$_2$ and KLaS$_2$ crystallize in the $\alpha$-NaFeO$_2$ structure in spacegroup $R\overline{3}m$ (Fig. 1). The lattice parameters of KCeS$_2$, determined at 100 K, are $a = b = 4.2225(2)$ Å and $c = 21.806(1)$ Å, in accordance with room temperature literature values [34].

# 3 Experimental and computational techniques

Static-field magnetization studies were performed using superconducting quantum interference device (SQUID) magnetometers from Quantum Design (MPMS-XL) and a physical property measurement system (PPMS), equipped with a vibrating sample magnetometer (VSM) option. Pulsed-field magnetization measurements up to 30 T were performed at the Hochfeld-Magnetlabor Dresden (HLD), using a compensated pickup-coil magnetometer in a $^4$He flow cryostat and a pulsed magnet with an inner bore of 20 mm, powered by a 1.44 MJ capacitor bank [36]. The background-corrected pulsed-field data were calibrated using our VSM measurements of another sample from the same batch.

ESR measurements were performed with a custom-built spectrometer based on a PNA-X network vector analyzer from Keysight Technologies which is used to generate and detect microwave radiation in the frequency range from 20 to 330 GHz. We employed a superconducting solenoid from Oxford Instruments equipped with a variable temperature insert and providing variable magnetic fields up to 16 T. The measurements were carried out in transmission geometry employing the Faraday configuration.

The crystal electric field (CEF) excitations in KCeS$_2$ have been probed by inelastic neutron scattering using the time-of-flight (TOF) neutron spectrometer MERLIN [37] at the ISIS neutron source of the Rutherford Appleton Laboratory.

Quantum chemical calculations were performed on a finite atomic cluster containing a

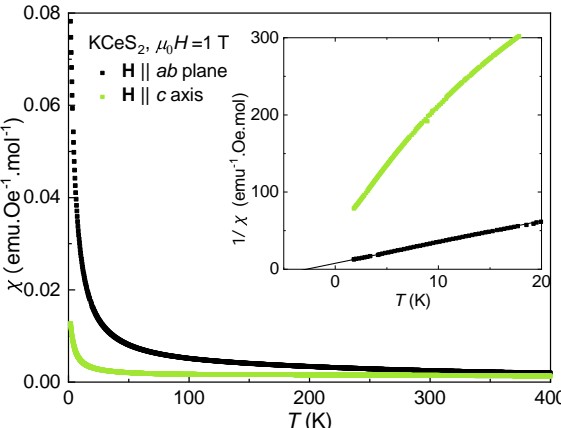

Figure 2: Temperature dependence of the magnetic susceptibility of $KCeS_2$ for magnetic fields in the basal plane $ab$ and along the $c$ axis. The measurements were performed with an applied magnetic field of 1 T. Insert: Inverse magnetic susceptibility $1/\chi$ as a function of temperature. The straight line indicates a Curie-Weiss fit.

central $CeS_6$ octahedron, the six Ce nearest-neighbors (NN's), along with twelve NN K ions around the central Ce site. The remaining part of the crystalline lattice was modeled as a large array of point charges optimized to reproduce the ionic Madelung field in the cluster region [38]. All quantum chemical calculations were carried out with the MOLPRO package [39]. In prior complete-active-space self-consistent-field (CASSCF) calculations [40], all seven $4f$ orbitals of the central Ce site were considered in the active space. The variational optimization was carried out for an average of all seven possible states associated with this manifold. The Ce $4f$ and S $3p$ electrons at the central octahedron were correlated in subsequent multireference configuration interaction (MRCI) calculations [40]. Spin-orbit interactions were introduced as described in Ref. [41] and implemented in the MOLPRO MRCI module. For the central Ce ion we employed energy-consistent relativistic pseudopotentials [42] and Gaussian-type valence basis functions of quadruple-zeta quality [43, 44] while for the S ligands we applied all-electron valence triple-zeta $((15s9p2d)/[5s4p2d])$ basis sets from the MOLPRO library [45, 46]. The K ions were described as total ion potentials [47]. Large-core pseudopotentials were also applied for the six Ce NN's [48].

The specific heat measurements were performed using a relaxation method with a PPMS equipped with a $^3$He refrigerator option. The crystals were mounted using a sapphire block with a 90° angle in order to achieve the configuration $\mathbf{H} \parallel a$ or $\mathbf{H} \parallel [1\bar{1}0]$. The background signal from the sample holder and sapphire block was separately measured and subtracted.

$\mu$SR experiments were performed at the PSI, Switzerland using the HAL-9500 spectrometer ($\pi$E3 beamline), equipped with a BlueFors vacuum-loaded cryogen-free dilution refrigerator (DR), and also at ISIS, U.K. using the MuSR instruments. For the measurements at ISIS, 300 mg of a powder sample, mixed with a small amount of GE varnish to ensure good thermal contact, was dispersed on a silver plate with a radius of 10 mm. The $\mu$SR data were analyzed with the free software packages MANTID [49] and MUSRFIT [50].

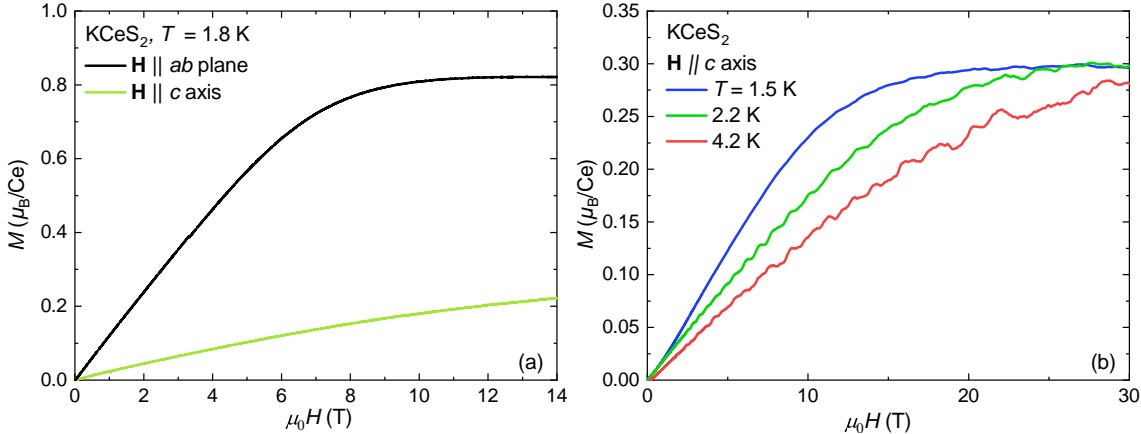

Figure 3: (a) Field dependence of the magnetization at 1.8 K, measured in static magnetic fields applied in the basal plane $ab$ and along the $c$ axis. (b) Field dependence of the magnetization along the $c$ axis, measured in pulsed magnetic fields up to 30 T at several temperatures.

## 4  Magnetization measurements

The temperature dependence of the magnetic susceptibility $\chi$ of KCeS$_2$ and its inverse $1/\chi$ are represented in Fig. 2 for magnetic fields applied in the basal plane $ab$ and along the $c$ axis. A strong easy-plane magnetic anisotropy can be observed in the whole temperature range up to 400 K and no signatures of magnetic phase transitions are observed down to 1.8 K. While several CEF levels, harboring different $g$-factor anisotropies [51], contribute to the magnetization in the high-temperature limit, the magnetization in the low-temperature limit $T < 20$ K reflects the properties of the well separated lowest energy doublet, as justified later via INS and quantum chemical calculations.

The in-plane magnetic susceptibility (**H** $\parallel ab$) follows the Curie-Weiss law below 20 K, yielding an effective moment of $\mu_{\text{eff},ab} = 1.7(1)$ $\mu_{\text{B}}$/Ce and a Curie-Weiss temperature of $\theta_{\text{CW},ab} = -2.8 \pm 1$ K. The Curie-Weiss temperature $\theta_{\text{CW},ab} = -2.8 \pm 1$ K indicates moderate antiferromagnetic interactions implying magnetic frustration, given the absence of long-range magnetic order down to 1.8 K. On the contrary magnetization measurements along the $c$ axis do not show any temperature interval with a clear realization of the Curie-Weiss law.

The magnetization at 1.8 K for static magnetic fields up to 14 T is presented in Fig. 3(a). For fields parallel to the $ab$ plane, a saturation moment of 0.82(2) $\mu_{\text{B}}$ is reached at about 12 T. In order to probe magnetic saturation in fields along the $c$ axis, magnetization measurements in pulsed magnetic fields were performed up to 30 T at several temperatures (Fig. 3(b)). A magnetic moment of 0.30(5) $\mu_{\text{B}}$/Ce was reached at about 20 T, implying a large $g$-factor anisotropy as detailed in the ESR section 5 below.

## 5  ESR measurements

An accurate determination of the $g$ tensor is important for a correct analysis of the static magnetic data because it enters in several quantities, such as the saturation magnetization and the Curie constant. Furthermore, the knowledge of the $g$ tensor facilitates the calculation of the crystal field parameters, and thus may be helpful for the analysis of the INS data (see

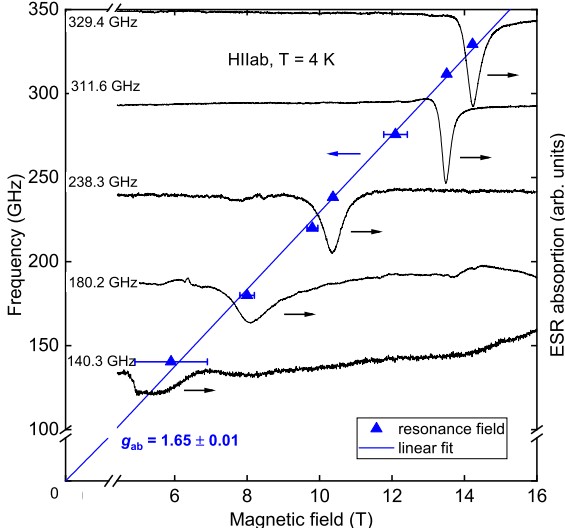

Figure 4: Exemplary ESR spectra (right vertical scale) and the frequency dependence of the resonance field $H_{\mathrm{res}}$ (left vertical scale) at $T = 4\,\mathrm{K}$ for the field applied perpendicular to the $c$ axis (solid triangles). The solid straight line is a fit to Eq. (1); see the text for details.

Section 6). The most precise means to obtain the elements of the $g$ tensor is the measurement of the ESR signal at several excitation frequencies $\nu$ for the magnetic field applied along the principal magnetic anisotropy axes $a$, $b$ and $c$ of a single-crystalline sample. Since in the paramagnetic state the resonance field $H_{\mathrm{res}}$ is related to $\nu$ via the resonance condition

$$\nu = g^{\mathrm{i}}\mu_{\mathrm{B}}\mu_0 H_{\mathrm{res}}^{\mathrm{i}}/h \quad (i = a, b, c), \tag{1}$$

the slope of the obtained $\nu$ vs. $H_{\mathrm{res}}$ dependence will be determined by the $i$-th component of the $g$ tensor corresponding to a specific direction of the applied magnetic field $H$. Here, $\mu_{\mathrm{B}}$, $\mu_0$, and $h$ are Bohr's magneton, the vacuum permeability, and Planck's constant, respectively.

Such a dependence, measured at $T = 4\,\mathrm{K}$ for $\mathbf{H} \parallel ab$ plane, together with exemplary ESR spectra are shown in Fig. 4. At high frequencies, a well-defined and relatively sharp ESR absorption line of $\mathrm{Ce}^{3+}$ ions is observed. With lowering of the excitation frequency $\nu$, the signal broadens and, consequently, its amplitude decreases so that the ESR signal cannot be detected anymore for $\nu < 140\,\mathrm{GHz}$. Such a broadening suggests enhanced spin fluctuations in small applied fields. These fluctuations are suppressed in the strong field limit due to polarization of the Ce spins, which manifests in the saturation of the static magnetization (see, Fig. 3). As expected, the resonance field $H_{\mathrm{res}}$ corresponding to the peak of the ESR signal depends linearly on $\nu$, and the corresponding fit using Eq. (1) yields the $g$-factor value $g_{ab} = 1.65 \pm 0.01$ for this field direction (Fig. 4).

Remarkably, for the field geometry $\mathbf{H} \parallel c$ axis, no ESR signal can be detected in the available frequency- and magnetic field range. Since for this field direction the spins are already partially polarized in fields of 14 - 16 T, the absence of a signal at high frequency due to strong spin fluctuations seems unlikely. A more plausible reason could be a much smaller $g$-factor value for this orientation ($g_c \ll g_{ab}$) which would require field strengths larger than 16 T for an observation of the resonance signal.

Indeed, a strong $g$-factor anisotropy has already been inferred from the static magnetic data. As is known, the ground state $^2F_{5/2}$ of a $\mathrm{Ce}^{3+}$ ion with the total angular momentum $J = 5/2$ ($J^z = \pm1/2, \pm3/2, \pm5/2$) is split in a crystal field into the three Kramers doublets

$|J, J^z\rangle$. In the particular case of an octahedral symmetry with a trigonal distortion, the effective $\tilde{S} = 1/2$ ground state doublet

$$| \pm \tilde{S}^z\rangle = \cos\alpha|5/2, \pm 1/2\rangle \pm \sin\alpha|5/2, \mp 5/2\rangle \tag{2}$$

is characterized in first-order theory by a uniaxially anisotropic $g$ tensor [52]:

$$g_c = g_L|(\cos^2\alpha - 5\sin^2\alpha)|, \tag{3}$$
$$g_{ab} = 3g_L\cos^2\alpha. \tag{4}$$

Here, $g_L = 6/7$ is the Lande factor of a free $Ce^{3+}$ ion and $\alpha$ is the so-called mixing angle which parameterizes the degree of distortion. The $g$-factor is isotropic for $\alpha = 41.8°$, corresponding to a regular octahedron and amounts to $g = g_c = g_{ab} = 1.43$, following Eqs. (3) and (4). This value obviously does not correspond to the ESR results on $KCeS_2$ with distorted $CeS_6$ octahedra. From the experimentally determined $g_{ab} = 1.65$ one obtains from Eq. (4) $\alpha = 36.8°$ which, according to Eq. (3), yields $g_c = 0.99$. This estimate sets an upper limit of $200 - 220$ GHz for the excitation frequency which can be used to detect an ESR signal for this orientation in the available field range. Considering a strong broadening of the line for $\nu < 200$ GHz observed for the other orientation, this could explain the non-observation of the ESR signal in $KCeS_2$ for $\mathbf{H} \parallel c$ axis.

From the obtained value of $g_{ab} = 1.65$, one can calculate the saturation magnetization of $KCeS_2$ for the in-plane direction of the applied magnetic field $M_{ab}^{sat} = g_{ab}\tilde{S} = 0.83\mu_B$. It nicely agrees with the value obtained from the static magnetization $M(H)$ measurements, evidencing that the full spin polarization is achieved with the in-plane field strength of $\sim 12$ T, and that the saturation values of $M$ are determined essentially by the $g$ tensor (see Section 4). Given this, the observed saturation magnetization for $\mathbf{H} \parallel c$ axis of $M_c^{sat} = 0.30\mu_B$ should correspond to $g_c = 0.6$. This value is significantly smaller than the above discussed estimate, which is not surprising considering the approximate character of the approach above. Indeed, the quantum chemistry calculations below yield a $g$ tensor which consistently explains both the ESR and the static magnetic data.

Finally, the earlier reported ESR results on a nonmagnetic analog $KLaS_2$ host single crystal doped with $5\%$ Ce should be briefly commented [53]. Since, due to a strong dilution, magnetic interactions between the $Ce^{3+}$ ions could be avoided, the ESR signal was detected at a relatively small excitation frequency of $\sim 18$ GHz and thus both components of the $g$ tensor were determined. The obtained values of $g_c = 0.47$ and $g_{ab} = 1.745$ differ from those for the concentrated stoichiometric $KCeS_2$, apparently due to a somewhat different local crystal field acting on Ce dopants in $KLaS_2$ as compared to the crystal field at the regular Ce sites in $KCeS_2$.

## 6 Inelastic neutron scattering

The data [54] presented in Fig. 5 were collected with an incident neutron energy $E_i = 131$ meV, using a powder sample with a mass of $\sim$2 g. At the base temperature $T = 5$ K, we observe two intense crystal-field excitations at 46.7 and 61.7 meV, characterized by a monotonically decaying intensity as a function of momentum transfer, $|\mathbf{Q}|$, in accordance with the magnetic form factor.

At the base temperature $T = 5$ K, the width of the CEF lines is limited by the experimental energy resolution, evidencing the absence of any considerable CEF randomness resulting from the site intermixing that was reported, for instance, in $YbMgGaO_4$ [55]. With increasing temperature, both lines gradually broaden, as can be seen in Fig. 5 (b), without any significant change in their integrated intensity. Due to the relatively high energy of the first excited

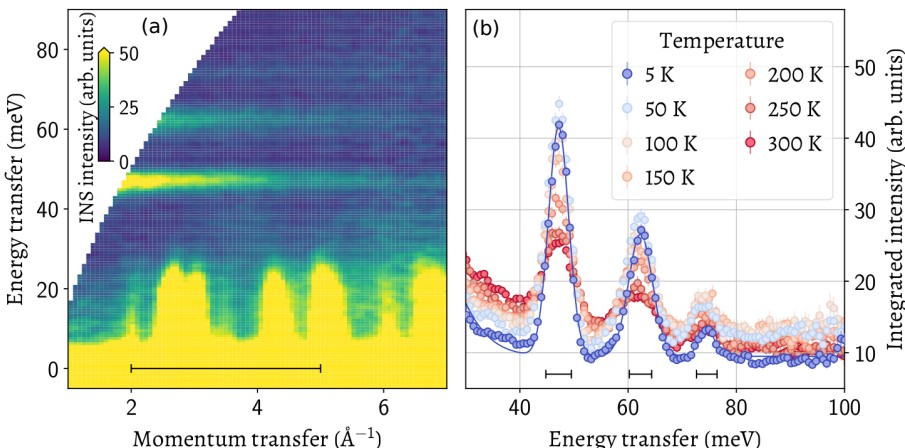

Figure 5: Crystal-electric-field excitations in $KCeS_2$, measured using TOF neutron spectroscopy. (a) Color map of the INS intensity at $T = 5$ meV, measured with an incident neutron energy $E_i = 131$ meV. (b) The energy spectrum, integrated over the $|\mathbf{Q}|$ range shown by the black horizontal bar in panel (a), presented for different temperatures from 5 to 300 K. The horizontal bars below the peaks show calculated energy resolution, indicating that the peak widths at base temperature are resolution-limited.

CEF state in comparison to room temperature, no additional transitions from the temperature-populated excited state could be observed up to 300 K. Thus, the INS spectrum evidences a well separated $\tilde{S} = 1/2$ ground state of $KCeS_2$. The experimental spectrum shows a weak additional peak around ~74 meV, yet its intensity is too small to draw any definite conclusion about its temperature dependence or the $|\mathbf{Q}|$-dependence of its form factor. Knowing that only two low-energy CEF transitions are expected for the $Ce^{3+}$ ion (which is confirmed by the first-principles calculation), the origin of this additional line is unclear. It can possibly originate from a minority of Ce ions in a different surrounding, for instance in the vicinity of crystal defects such as stacking faults that are typical for the layered delafossite structure [56–58]. Similar additional CEF lines have been previously observed, for instance, in the isostructural $NaYbO_2$ [25], $NaYbS_2$ [19], $NaYbSe_2$ [28], and in the Ce pyrochlore $Ce_2Zr_2O_7$ [59].

## 7 Quantum chemical calculations

For additional insights into the underlying electronic structure, we performed embedded-cluster quantum chemical computations. Ce-ion $4f$-shell crystal-field splittings as obtained by CASSCF and MRCI computations without spin-orbit coupling are listed in Table 1. The $D_{3d}$ environment splits the $4f$ manifold into two groups of doubly degenerate levels ($E_u$) and three non-degenerate states (one $A_{1u}$ and two $A_{2u}$ states) and our results indicate substantial crystal-field effects. Results of CASSCF and MRCI calculations accounting for spin-orbit interactions are listed in Table 2. The six-fold degeneracy of the free-ion $^2F_{5/2}$ term is also lifted due to the anisotropic surroundings to yield a set of three Kramers doublets (two $\Gamma_6$ and one $\Gamma_4 + \Gamma_5$ [60]) in the lower-energy part of the spectrum.

The excitation energies computed by SO-MRCI for the lower two excited states, 50 and 61 meV (see Table 2), are in reasonable agreement (better than 10%) with the excitations observed at 46.7 and 61.7 meV in the INS spectra. The computations also indicate a highly

Table 1: CASSCF and MRCI results for the $f$-shell single-electron levels without spin-orbit coupling in $KCeS_2$. The states are labeled according to notations in $D_{3d}$ point-group symmetry [61].

| $Ce^{3+}$ $4f^1$ CF states | Relative energies (meV) | |
| --- | --- | --- |
| | CASSCF | MRCI |
| $^2A_{2u}$ | 0.0 | 0.0 |
| $^2E_u$ | 32.0 | 32.0 |
| $^2A_{1u}$ | 49.0 | 46.0 |
| $^2E_u$ | 93.0 | 91.0 |
| $^2A_{2u}$ | 127.0 | 124.0 |

Table 2: $Ce^{3+}$ $4f^1$ electronic structure as obtained by SO-CASSCF and SO-MRCI for $KCeS_2$. Units of meV and notations for trigonal symmetry are used [60].

| $4f^1$ spin orbit states | Relative energies (meV) | |
| --- | --- | --- |
| | SO-CASSCF | SO-MRCI |
| $\Gamma_6$ | 0.0 | 0.0 |
| $\Gamma_4 + \Gamma_5$ | 51.0 | 50.0 |
| $\Gamma_6$ | 64.0 | 61.0 |
| $\Gamma_6$ | 241.0 | 242.0 |
| $\Gamma_6$ | 279.0 | 279.0 |
| $\Gamma_4 + \Gamma_5$ | 299.0 | 297.0 |
| $\Gamma_6$ | 331.0 | 329.0 |

anisotropic ground-state $g$ tensor with $g_{ab} = 1.67$ and $g_c = 0.58$, in good agreement with ESR data ($g_{ab} = 1.65$). Corroborated with the magnetization, ESR, and INS results, the *ab initio* calculations unequivocally evidence a comprehensive picture: the $Ce^{3+}$ $4f^1$ ground-state term is well separated from higher-lying $4f^1$ Kramers doublets in $KCeS_2$ and is associated with strong easy-plane $g$-factor anisotropy.

# 8 Specific heat measurements

The specific heat of $KCeS_2$ as a function of temperature down to 0.36 K is presented in Fig. 6 in zero field and in magnetic fields applied in the basal plane $ab$. The zero-field specific heat shows a sharp second-order phase transition at $T_N = 0.38(1)$ K. This ordering temperature corresponds to a frustration ratio $f = \theta_{CW}/T_N$ of 7.4 indicating strong magnetic frustration.

The temperature dependence of the specific heat in applied magnetic fields within the basal plane $ab$ strongly depends on the exact direction of the magnetic field. This in-plane

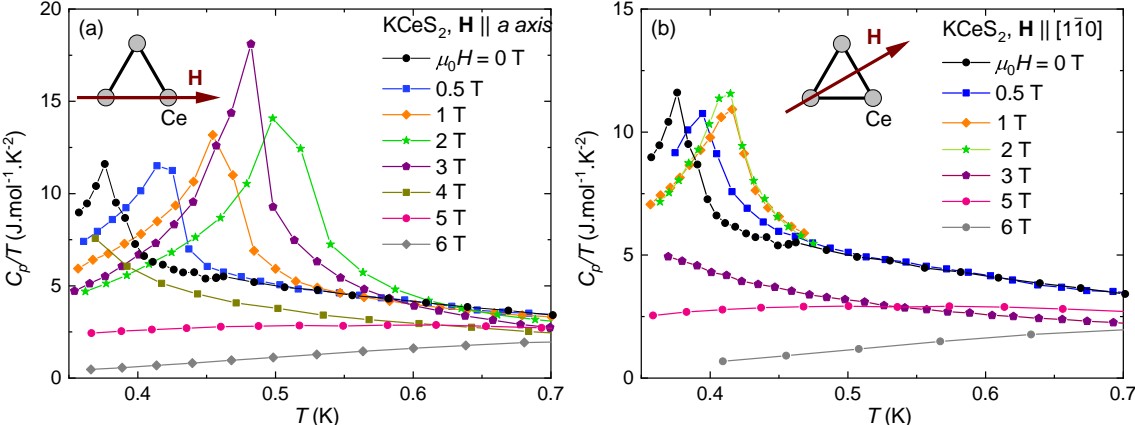

Figure 6: Specific heat divided by temperature $C_p/T$ of KCeS$_2$ as a function of temperature in the range 0.35 to 0.70 K for different magnetic fields applied (a) along the $a$ axis and (b) in the $ab$ plane parallel $[1\bar{1}0]$. The insets indicate the direction of the magnetic field with respect to the triangular lattice built by the Ce atoms.

anisotropy harbors a periodicity of 60° (data not shown), in agreement with the $R\bar{3}m$ symmetry observed T >100 K by x-ray diffraction (see Fig. 1). The temperature dependence of the in-plane specific heat of KCeS$_2$ for fields ‖ $a$ and ‖ $[1\bar{1}0]$ is presented in Fig. 6 (a) and (b), respectively. For both field directions, the Néel temperature first increases with increasing magnetic field, and decreases for $\mu_0 H \gtrsim$ 1.5–2 T.

In order to determine the magnetic contribution to the specific heat of KCeS$_2$, the lattice contribution needs to be subtracted. It has been approximated by the specific heat of the nonmagnetic structural analog compound KLaS$_2$. $C_p/T$ of KCeS$_2$ and KLaS$_2$ and the corresponding magnetic contribution $C_{p,\text{mag}}/T$ of KCeS$_2$ are presented in Fig. 7(a) as a function of temperature up to 20 K and in magnetic fields up to 9 T applied along $[1\bar{1}0]$. The correction by the Lindemann scaling factor to account for the difference of molar mass and molar volume between the two compounds [62] was included despite the correction being very small. The magnetic contribution to the specific heat of KCeS$_2$ in the absence of external magnetic fields collapses rapidly upon warming and can be resolved only up to about 10 K, indicating a paramagnetic state with rather weak magnetic correlations and a negligible influence of higher energy CEF levels between 10 K and 20 K.

For $\mu_0 H > 5$ T the low-temperature specific heat $C_{p,\text{mag}}/T$ undergoes a broad maximum as a function of temperature, shifting to higher temperature with increasing magnetic field. This broad maximum corresponds to the onset of ferromagnetic correlations by entering the magnetically saturated regime, and reaches 2 K at 9 T, in agreement with the observation of magnetic saturation, see Fig. 3. A similar scenario applies for the specific heat data for **H** ‖ $a$ (not shown).

The magnetic entropy $S_{\text{mag}}(T) = \int C_{p,\text{mag}}/T\,dT$ is presented in Fig. 7 (b) as a function of temperature. At the highest magnetic field $\mu_0 H = 9$ T, the collapse of $C_{p,\text{mag}}/T$ at low temperature indicates a vanishing magnetic entropy by entering the saturated regime, for which we assumed $S(T = 0.35 \text{ K}, \mu_0 H = 9 \text{ T}) \simeq 0$. In order to get an estimate of the magnetic entropy for the other magnetic fields on an absolute scale, we use the Maxwell relation $\partial S / \partial (\mu_0 H) = \partial M / \partial T$. The magnetization at $T = 20$ K remains far from saturation at least until $\mu_0 H = 9$ T, and can thus reasonably be considered as linear in field. Then the magnetic entropy at $T_0 = 20$ K and a magnetic field $H_2$ can be estimated from the magnetic entropy at another magnetic field $H_1$ via the equation:

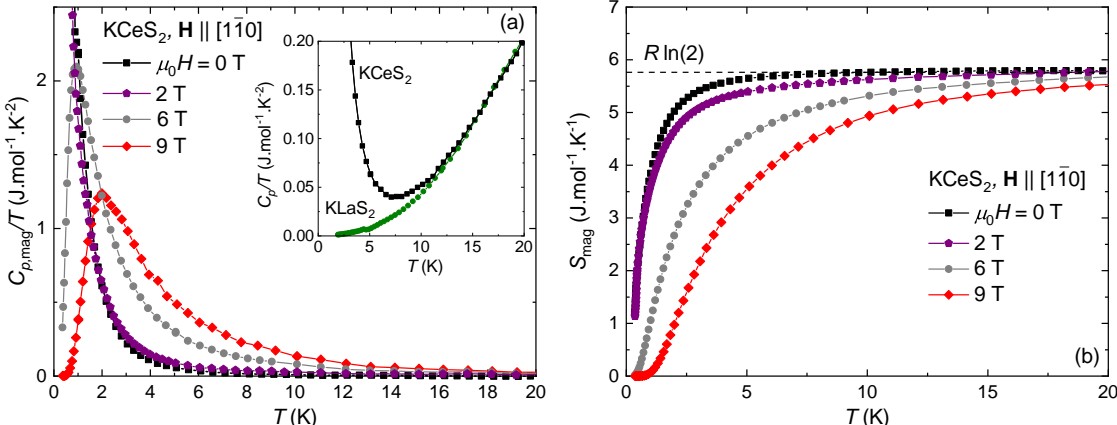

Figure 7: (a) Magnetic contibution to the specific heat divided by temperature $C_{p,\mathrm{mag}}/T$ of KCeS$_2$ as a function of temperature in the range 0.35 to 20 K for different magnetic fields applied in the $ab$ plane parallel $[1\bar{1}0]$. The inset shows the specific heat divided by temperature $C_p/T$ of KCeS$_2$ and the nonmagnetic structural analog compound KLaS$_2$. These data were used to subtract the phonon contribution to the specific heat, in order to obtain the magnetic contribution $C_{p,\mathrm{mag}}/T$. (b) Magnetic entropy $S_{\mathrm{mag}}(T)$ of KCeS$_2$ in different magnetic fields applied in the $ab$ plane parallel $[1\bar{1}0]$.

$$S_{\mathrm{mag}}(T_0, H_2) = S_{\mathrm{mag}}(T_0, H_1) + \frac{\mu_0}{2}\left(\frac{d\chi}{dT}\right)_{T=T_0}(H_2^2 - H_1^2). \tag{5}$$

The resulting magnetic entropy, represented in Fig. 7 (b), saturates in the absence of an external magnetic field and in magnetic fields applied along $[1\bar{1}0]$. It saturates around $R\ln(2)$ for the different magnetic fields confirming the occurrence of $\tilde{S} = 1/2$ effective spins for temperatures below $T = 20$ K.

The maximum in $C_p/T$ at the Néel temperature at finite magnetic field occuring for both field directions may be a signature for a field-induced up-up-down phase [5, 63], as observed in $S = 1/2$ TLAF, such as in Ba$_3$CoSb$_2$O$_9$ [64] and in the Yb-based TLAF NaYbO$_2$ [22–24] and NaYbSe$_2$ [27]. However, an oblique phase may also occur as proposed for NaYbS$_2$ [65] and NaYbSe$_2$ [27] in applied magnetic fields. Contrary to other rare-earth based delafossites, the remarkable feature of KCeS$_2$ is the anisotropy observed for the magnetic ordering temperature within the basal plane. While for $\mathbf{H} \parallel a$ the ordering temperature undergoes a maximum at 0.51 K for $\mu_0 H = 2$ T, for $\mathbf{H} \parallel [1\bar{1}0]$ this maximum is reduced to 0.41 K at $\mu_0 H = 1.5$ T. The corresponding magnetic field-temperature phase diagram is shown in Fig. 8 for the two different in-plane directions.

# 9 $\mu$SR experiments

In order to investigate the static and dynamic properties of the magnetic ground state of KCeS$_2$ we have performed $\mu$SR experiments in the temperature range 0.08–2.5 K in zero field (ZF). Representative ZF-$\mu$SR asymmetry spectra at different temperatures are shown in Fig. 9(a). Below 0.4 K the spectra display characteristic signals from static bulk magnetism. Although the spectra do not display any spontaneous coherent oscillations in the studied temperature range down to 0.08 K in the time range up to 20 $\mu$s, they show a strong temperature dependent relaxation of the muon spin polarization. A recovery of 1/3 of the muon spin polarization at

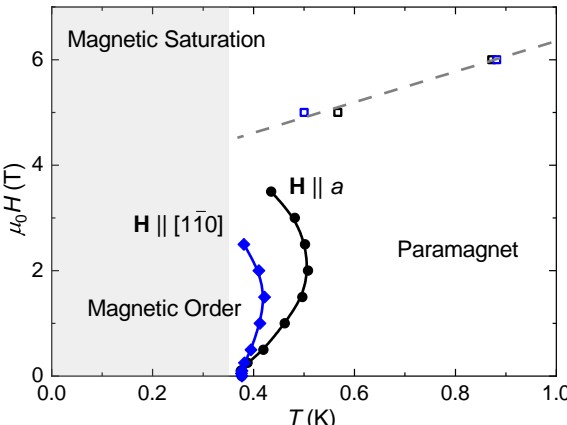

Figure 8: Magnetic field-temperature phase diagram of KCeS$_2$ from specific heat measurements. The black and blue points correspond to the phase boundaries in magnetic fields applied along the $a$ axis and parallel $[1\bar{1}0]$, respectively. While full symbols and solid lines stand for second-order magnetic phase transitions, the open symbols and dashed lines indicate the broad crossover from the paramagnetic state towards magnetic saturation, as observed from specific heat measurements. The gray area is the region not investigated in this study.

late times expected for the case of a random distribution of static internal magnetic fields at the muon site is not observed. The asymmetry at late times reduces almost to zero. The muon spin depolarizes faster at low temperatures, indicating the presence of a broad disordered static magnetic field distribution at one muon stopping site.

To describe the zero-field data adequately, a sum of two relaxation functions is required, with an individual probability of about 50%, which was assumed to be temperature independent. This suggests the same population of two magnetically different muon sites. The ZF-$\mu$SR spectra can be adequately described by the following function in the whole temperature range studied:

$$A(t) = A_1 e^{-\lambda_1 t} + A_2 e^{-\lambda_2 t} + B_{\text{bg}}, \tag{6}$$

where $A_1$, $A_2$ represent the initial asymmetry, and $B_{\text{bg}} \sim 0$ is the constant background, predominantly caused by the muons stopped outside the sample. $\lambda_1$ and $\lambda_2$ are the muon relaxation rates. Allowing a temperature dependence of the ratio $A_1/A_2$ affects the relaxation rate values but the temperature dependence remains qualitatively unchanged. The observations are similar to the findings for NaYbO$_2$ [23]. For NaYbO$_2$ two possible muon sites with the same population but different distances to the Yb ions were proposed. Given that KCeS$_2$ and NaYbO$_2$ adapt the same crystal structure, one can expect a similar situation for KCeS$_2$.

Fig. 9(b) and (c) show the muon spin relaxation rates $\lambda_1$ and $\lambda_2$ respectively, as a function of temperature down to 50 mK. We find two relaxation rates with very different absolute values, clearly revealing that muons at two different sites are coupled differently to the Ce$^{3+}$ magnetic moments. One relaxation rate ($\lambda_1$), whose values are above 10 $\mu s^{-1}$, constantly increases upon lowering temperature (see Fig. 9(b)). Below 0.4 K $\lambda_1$ shows a stronger increase until it settles down at relatively high relaxation values at lowest temperatures. $\lambda_2$ exhibits a weak maximum at 0.4 K and a constant value for $T \rightarrow 0$ (Fig. 9(c)). In a magnetically ordered system a peak of the muon spin relaxation rate at the magnetic ordering temperature is typically caused by slow magnetic field fluctuations due to the divergence of the spin correlation time.

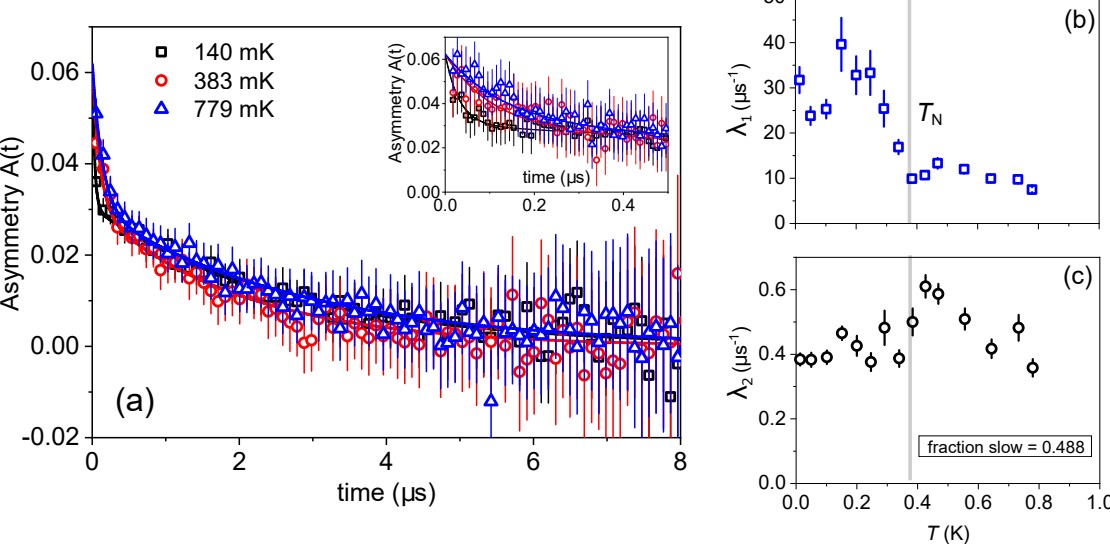

Figure 9: (a) ZF $\mu$SR time spectra collected at different temperatures. Solid lines represent fits as described in the main text. (b) and (c) Temperature dependence of the muon spin relaxation rates (b) $\lambda_1$ and (c) $\lambda_2$ for the ZF-time spectra, stemming from two different muon sites in the sample. Vertical shaded lines indicate the antiferromagnetic ordering temperature.

In summary the ZF muon spin relaxation experiments support the specific heat measurements, implying the presence of a magnetically ordered state below 0.38 K in KCeS$_2$. They demonstrate that at zero-field KCeS$_2$ significantly differs from the other Ce- and Yb-based TLAF. In NaYbO$_2$ and NaYbS$_2$ no evidence of bulk static magnetism was found [19, 23, 26].

## 10  Discussion

KCeS$_2$ provides an example of a magnetically ordered $\tilde{S} = 1/2$ TLAF with the delafossite structure, in contrast to the putative QSL candidates NaYbO$_2$ [21–24], NaYbS$_2$ [19, 21, 26], NaYbSe$_2$ [21, 27], KYbS$_2$ [29], CsYbSe$_2$ [30] and CsCeSe$_2$ [30]. Specific heat measurements of each of these QSL candidates indicate a broad maximum of $C_p/T$ around the temperature corresponding to the possible formation of the spin liquid state [19, 22, 27, 29, 30]. This broad maximum is absent in KCeS$_2$ and replaced by magnetic ordering below $T_N = 0.38$ K. However, it should be emphasized, that despite a different magnetic ground state, KCeS$_2$ shows strong similarities with the QSL candidates NaYbO$_2$, NaYbS$_2$ and NaYbSe$_2$ in terms of a well separated $\tilde{S} = 1/2$ state and a strong easy-plane $g$-factor anisotropy [19, 24, 27, 66, 67]. Thus the open question, which is raised by the present publication is: why does KCeS$_2$ order magnetically in contrast to many other $\tilde{S} = 1/2$ rare-earth based delafossites?

The origin of the QSL state in Yb- and Ce-based delafossites remains unclear and under debate. Anisotropic first neighbor magnetic interactions $J_{z\pm}$ or second nearest-neighbor interactions $J_2$ may be tuning parameters to drive the TLAF from a magnetically ordered ground state to a quantum spin liquid state [9, 13–16, 16, 17]. Such magnetic interactions must be very sensitive to the details of the crystal structures such as the bond angles, as evidenced for the case of Yb-based magnets by effective superexchange models [32]. Thus the difference of the magnetic ground state could simply come from slight differences of the cell parameters between KCeS$_2$ ($a = b = 4.2225(2)$ Å and $c = 21.806(1)$ Å) and for example KYbS$_2$ ($a = b = $

3.968 Å and $c = 21.841(2)$ Å [68] or CsCeSe$_2$ ($a = b = 4.4033$ Å and $c = 24.984$ Å) [30].

Two other examples of magnetically ordered Ce-based TLAF were previously reported: CeZn$_3$P$_3$ with $T_N$=0.8 K [69] and CeCd$_3$P$_3$ with $T_N$=0.41 K [70]. They both crystallize in the ScAl$_3$C$_3$ structure with well separated CeP$_2$ layers similar to the CeS$_2$ layer of KCeS$_2$. In addition, a very strong similarity of the magnetic field-temperature phase diagram for CeCd$_3$P$_3$ and KCeS$_2$ should be noticed [70], suggesting similar magnetic ground states and/or field-induced magnetic states. However, magnetic order in CeCd$_3$P$_3$ may be favored by the possible metallic behavior of this compound, which remains under debate [70, 71].

A rather unique property of KCeS$_2$ is the in-plane anisotropy of the magnetic field-temperature phase diagram. Such anisotropy effects have not been reported in the Yb-based TLAF NaYbS$_2$, NaYbSe$_2$ and CsYbSe$_2$ [27,65,72], nor in the Ce-based TLAF CeCd$_3$P$_3$ [70]. This in-plane anisotropy provides constraints on the spin direction in the magnetically ordered state and thus also on the actual ground state or field-induced magnetic state of KCeS$_2$. In the case of a collinear up-up-down phase, the measured anisotropy of the field dependence of the ordering temperature would indicate that the magnetic moments in this phase point preferably along a nearest-neighbor bond. This anisotropy gives constraints on the properties of the anisotropic magnetic interactions, which are proposed to occur in rare-earth based TLAF [13, 32]. Indeed a negative anisotropic interaction $J_{\pm\pm}$ in the Hamiltonian proposed in Ref. [13], even smaller than $J_{\pm}$ would favor a collinear up-up-down phase along the Ce-Ce bond compared to a collinear up-up-down phase transverse to the Ce-Ce bond and would thus be in principle agreement with the observed in-plane magnetic anisotropy. Such a scenario needs a microscopic confirmation of the occurrence of the common up-up-down phase in KCeS$_2$ in applied magnetic fields. It should be emphasized, that Quantum Monte Carlo and DMRG studies have shown that the anisotropic magnetic interaction $J_{\pm\pm}$ favors magnetically ordered states [15–17] and thus strong $J_{\pm\pm}$ interactions might also be the origin of the absence of a QSL state in KCeS$_2$.

# 11 Conclusion

A magnetically ordered ground state was identified and characterized in the Ce-based TLAF KCeS$_2$ below $T_N = 0.38$ K by means of specific heat and $\mu$SR experiments. In addition, the CEF scheme was drawn from CASSCF and MRCI calculations and confirmed by INS experiments, indicating the first two excited states at 46.7 and 61.7 meV. Magnetization measurements, ESR, and quantum chemical electronic-structure computations evidence, in good agreement, a strong easy-plane $g$-factor anisotropy. An anisotropy within the basal plane of the field dependent specific heat and the magnetic field-temperature phase diagram was further detected and might be a signature of anisotropic magnetic interactions.

The magnetically ordered ground state of KCeS$_2$ occurs despite strong similarities in terms of crystal structure, CEFs schemes and magnetic anisotropy with several recently reported QSL candidates such as NaYbS$_2$. Further microscopic measurements such as neutron diffraction to determine the microscopic spin alignment in the ground state of KCeS$_2$ and further theoretical efforts are needed to fully characterize the magnetic order in KCeS$_2$ and more generally to clarify the conditions for the realization of QSLs in rare-earth-based TLAF.

# Acknowledgements

Insightful discussions with M. Baenitz, M. Vojta as well as technical assistance by S. Gass, J. Scheiter, Y. Skourski and D. Gorbunov are acknowledged. We acknowledge financial sup-

port from the German Research Foundation (DFG) through the Collaborative Research Center SFB 1143 (project-id 247310070), the Würzburg-Dresden Cluster of Excellence on Complexity and Topology in Quantum Matter – *ct.qmat* (EXC 2147, project-id 390858490) and project HO-4427/3, as well as the support from the European Union's Horizon 2020 research and innovation programme under the Marie Skłodowska-Curie grant agreement No 796048 and from the HLD at HZDR, a member of the European Magnetic Field Laboratory (EMFL). Z. Z. and L. H. thank U. Nitzsche for technical assistance.

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
