# Peer review of "Long-range magnetic order in the ${\tilde S}=1/2$ triangular lattice antiferromagnet KCeS$_2$"

_SciPost Physics, doi:SciPost Phys. 9, 041 (2020)_

## Round 2 · Referee Report · Anonymous (Referee 1) · 2020-6-19

Strengths

This is a very important, timely, thorough, and comprehensive study.

Report

This is a very important, timely, thorough, and comprehensive study of one of the new material of an emergent family of the rare-earth-based quantum magnets with a frustrated geometry. Studies of such materials are important to elucidate the mutual effect of the geometric frustration and strong spin-orbit couplings and to understand the nature of the quantum-disordered and unusual ordered states this combination can yield.

The present study focuses on a particular representative of the triangular-lattice family of these interesting group of materials. Here, the Kramers ion is Ce3+, which offers a variety to other studies that were mostly based on Yb3+ material.
I find it rather remarkable, and very much commendable, that the authors combined so many different measurements and techniques into one comprehensive picture.

While I understand that the temperatures are prohibitive for some of the measurements (like C(T)), given the fact that there seems to be no additional phases vs field (like plateaus), and may be the overall phase diagram of the model that is known from theory, where KCeS2 most likely belongs to according to the authors? Which phase? An extra comment on that would be appreciated.

Requested changes

1) Fig. 8 has an unnecessary "(a)" in the caption. 2) One improvement suggested: Although it will be clear for most of the readership of this paper, I strongly suggest to the authors to have a figure with the plane of Ce layers with "a-direction" and "[1,-1,0] direction" for the field explicitly identified.

  • validity: top
  • significance: high
  • originality: high
  • clarity: top
  • formatting: excellent
  • grammar: excellent

Author:  Gaël Bastien  on 2020-07-22  [id 896]

(in reply to Report 1 on 2020-06-19)

We thank the referee for careful reading of the paper, for her/his positive reports and recommendations for the publication in Scipost Physics.

Some general phase diagrams for the triangular lattice antiferromagnets were proposed from theory including further nearest neighbor interactions and/or anisotropic magnetic interactions (see references 10,11,14-18). These theoretical phase diagrams include several antiferromagnetically ordered phases such as the 120° spin-ordered phase and stripy magnetic phases. However, we could not estimate the magnetic interactions of KCeS2 in the present work and thus prefer not to speculate where KCeS2 lays on these theoretical phase diagrams.

We agree with the suggestions concerning figures 6 and 8 and changed our manuscript accordingly.

---

## Round 2 · Referee Report · Anonymous (Referee 2) · 2020-7-5

Strengths

  1. The quality of sample is very good.
  2. The characterization of sample looks excellent.

Weaknesses

  1. Little is reported on quantum magnetic properties of KCeS2 because the exchange constant are very small and the ordering temperature is very low (TN=0.38 K).

Report

Report_SciPost_Phys_Bastien

Quantum many-body effects of quantum triangular lattice antiferromagnets (TLAF) are of great interest. This manuscript reports the magnetic properties of spin-1/2 TLAF KCeS2 with the delafossite structure. The authors succeeded in growing high quality small single crystals. They conducted characterization of KCeS2 using magnetization, X-ray diffraction, inelastic neutron scattering, ESR, specific heat and μSR measurements combined with quantum chemistry calculation. The authors confirmed that the lowest electronic level is described by effective spin-1/2 from the crystal field excitation spectrum. They evaluated g-factors for two different field directions from ESR and magnetization data and analyzed on the basis of crystal field theory and quantum chemistry calculation. This is essential to discuss the magnetic properties of magnet composed of Ce3+. The authors detected magnetic phase transitions at TN=0.38 K via specific heat and μSR measurements. The specific heat anomaly at the TN looks fairly sharp, which indicates high quality of sample and the absence of lattice disorder. Note that many so called quantum spin liquid (QSL) candidates have been reported, although the origin of the absence of is not attributed to the QSL but to the lattice disorder owing to the intersite mixing of atoms and charge disorder. Unfortunately, little is reported on quantum magnetic properties of KCeS2 because the exchange constant are very small and the ordering temperature is very low (TN=0.38 K). Anyway, this is a good piece of work, and thus I recommend publication in SciPost Physics.
  • validity: high
  • significance: good
  • originality: high
  • clarity: high
  • formatting: excellent
  • grammar: excellent

Author:  Gaël Bastien  on 2020-07-22  [id 897]

(in reply to Report 2 on 2020-07-05)

We thank the referee for her/his careful reading of the paper, for her/his positive reports and recommendations for the publication in Scipost Physics.

As pointed out by the referee, quantum magnetic fluctuations are expected to play a role due to the small exchange constants and ordering temperatures of KCeS2. We agree with the referee, that this is an interesting point, but which is not addressed in the present paper. Our work, however, will probably motivate future inelastic neutron scattering experiments at low temperature to measure the magnetic excitations in the ground state, and which is mentioned in the outlook of our manuscript.

---

## Round 3 · Referee Report · Anonymous (Referee 1) · 2020-7-23

Report

The small changes requested in the previous report are implemented.
The paper should be accepted as is.

---

## Round 3 · Referee Report · Anonymous (Referee 3) · 2020-7-23

Strengths

1- extensive study of new triangular lattice antiferromagnet, KCeS2, member of a group of interesting materials for spin liquid physics

2- range of experimental and theoretical tools used is very large

Weaknesses

1- authors cannot derive firm conclusions on controlling element of spin liquid vs. magnetic ground state

Report

In the present work the authors present an extensive study of a new triangular lattice antiferromagnet, KCeS2, member of a group of interesting materials studied intensively for spin liquid physics. From a thermodynamic and microscopic study they firmly establish the ground state of the f-electron carrying element Ce, which is in a well-defined S=1/2 state. Moreover, again from a (different) thermodynamic and microscopic study they also establish that the material undergoes a transition into some kind of antiferromagnetic order at low temperatures and establish the basic features of the magnetic phase diagram. Taking all data together, the authors conclude that the material is a triangular frustrated magnet, which different from closely related materials does not have a quantum spin liquid ground state, but rather the mentioned antiferromagnetic one. Unfortunately, the authors are not able to derive firm conclusions as to the controlling element driving some of these triangular magnets into a spin liquid, others into a magnetic ground state. One might argue, the authors found just "another magnet".

All in all, the paper is well-written and represents a very thorough experimental and theoretical characterization of the studied material. It contributes to the present discussions of these types of materials and meets the criterion of a novel and inclusive approach from different research areas to understand new materials. I therefore recommend the paper for publication.

---

## Editorial Decision

published